# GANs Can Play Lottery Tickets Too

**Xuxi Chen[1]\*, Zhenyu Zhang[1]\*, Yongduo Sui[1], Tianlong Chen[2]**
[1]University of Science and Technology of China, [2]University of Texas at Austin
{chanyh,zzy19969,syd2019}@mail.ustc.edu.cn, tianlong.chen@utexas.edu

## ABSTRACT

Deep generative adversarial networks (GANs) have gained growing popularity in numerous scenarios, while usually suffer from high parameter complexities for resource-constrained real-world applications. However, the compression of GANs has less been explored. A few works show that heuristically applying compression techniques normally leads to unsatisfactory results, due to the notorious training instability of GANs. In parallel, the *lottery ticket hypothesis* shows prevailing success on discriminative models, in locating sparse *matching subnetworks* capable of training in isolation to full model performance. In this work, we for the first time study the existence of such trainable matching subnetworks in deep GANs. For a range of GANs, we certainly find matching subnetworks at 67%-74% sparsity. We observe that with or without pruning discriminator has a minor effect on the existence and quality of matching subnetworks, while the initialization weights used in the discriminator plays a significant role. We then show the powerful transferability of these subnetworks to unseen tasks. Furthermore, extensive experimental results demonstrate that our found subnetworks substantially outperform previous state-of-the-art GAN compression approaches in both image generation (e.g. SNGAN) and image-to-image translation GANs (e.g. CycleGAN). Codes available at https://github.com/VITA-Group/GAN-LTH.

## 1 INTRODUCTION

Generative adversarial networks (GANs) have been successfully applied to many fields like image translation (Jing et al., 2019; Isola et al., 2017; Liu & Tuzel, 2016; Shrivastava et al., 2017; Zhu et al., 2017) and image generation (Miyato et al., 2018; Radford et al., 2016; Gulrajani et al., 2017; Arjovsky et al., 2017). However, they are often heavily parameterized and often require intensive calculation at the training and inference phase. Network compressing techniques (LeCun et al., 1990; Wang et al., 2019; 2020b; Li et al., 2020) can be of help at inference by reducing the number of parameters or usage of memory; nonetheless, they can not save computational burden at no cost. Although they strive to maintain the performance after compressing the model, a non-negligible drop in generative capacity is usually observed. A question is raised:

*Is there any way to compress a GAN model while preserving or even improving its performance?*

The lottery ticket hypothesis (LTH) (Frankle & Carbin, 2019) provides positive answers with *matching subnetworks* (Chen et al., 2020b). It states that there exist matching subnetworks in dense models that can be trained to reach a comparable test accuracy to the full model within similar training iterations. The hypothesis has successfully shown its success in various fields (Yu et al., 2020; Renda et al., 2020; Chen et al., 2020b), and its property has been studied widely (Malach et al., 2020; Pensia et al., 2020; Elesedy et al., 2020). However, it is never introduced to GANs, and therefore the presence of matching subnetworks in generative adversarial networks still remains mysterious.

To address this gap in the literature, we investigate the lottery ticket hypothesis in GANs. One most critical challenge of extending LTH in GANs emerges: how to deal with the discriminator while compressing the generator, including *(i) whether prunes the discriminator simultaneously* and *(ii) what initialization should be adopted by discriminators during the re-training?* Previous GAN compression methods (Shu et al., 2019; Wang et al., 2019; Li et al., 2020; Wang et al., 2020b) prune the generator model only since they aim at reducing parameters in the inference stage. The effect of

---

*Equal Contribution.

pruning the discriminator has never been studied by these works, which is unnecessary for them but possibly essential in finding matching subnetworks. It is because that finding matching subnetworks involves re-training the whole GAN network, in which an imbalance in generative and discriminative power could result in degraded training results. For the same reason, the disequilibrium between initialization used in generators and discriminators incurs severe training instability and unsatisfactory results.

Another attractive property of LTH is the powerful transferability of located matching subnetworks. Although it has been well studied in discriminative models (Mehta, 2019; Morcos et al., 2019; Chen et al., 2020b), an in-depth understanding of transfer learning in GAN tickets is still missing. In this work, we not only show whether the sparse matching subnetworks in GANs can transfer across multiple datasets but also study what initialization benefits more to the transferability.

To convert parameter efficiency of LTH into the advantage of computational saving, we also utilize channel pruning (He et al., 2017) to find the structural matching subnetworks of GANs, which enjoys the bonus of accelerated training and inference. Our contributions can be summarized in the following four aspects:

- Using unstructured magnitude pruning, we identify matching subnetworks at 74% sparsity in SNGAN (Miyato et al., 2018) and 67% in CycleGAN (Zhu et al., 2017). The matching subnetworks in GANs exist no matter whether pruning discriminators, while the initialization weights used in the discriminator are crucial.

- We show that the matching subnetworks found by iterative magnitude pruning outperform subnetworks extracted by randomly pruning and random initialization in terms of extreme sparsity and performance. To fully exploit the trained discriminator, we using the dense discriminator as a distillation source and further improve the quality of winning tickets.

- We demonstrate that the found subnetworks in GANs transfer well across diverse generative tasks.

- The matching subnetworks found by channel pruning surpass previous state-of-the-art GAN compression methods (*i.e.*, GAN Slimming (Wang et al., 2020b)) in both efficiency and performance.

## 2 RELATED WORK

**GAN Compression** Generative adversarial networks (GANs) have succeeded in computer vision fields, for example, image generation and translation. One significant drawback of the generative models is the high computational cost of the models' complex structure. A wide range of neural network compression techniques has been applied to generative models to address this problem. There are several categories of compression techniques, including pruning (removing some parameters), quantization (reducing the bit width), and distillation. Shu et al. (2019) proposed a channel pruning method for CycleGAN by using a co-evolution algorithm. Wang et al. (2019) proposed a quantization method for GANs based on the EM algorithm. Li et al. (2020) used a distillation method to transfer knowledge of the dense to the compressed model. Recently Wang et al. (2020b) proposed a GAN compression framework, GAN slimming, that integrated the above three mainstream compression techniques into a unified form. Previous works on GAN pruning usually aim at finding a sparse structure of the trained generator model for faster inference speed, while we are focusing on finding trainable structures of GANs following the lottery ticket hypothesis. Moreover, in existing GAN compression methods, only the generator is pruned, which could undermine the performance of re-training since the left-out discriminator may have a stronger computational ability than the pruned generator and therefore cause a degraded result due to the imparity of these two models.

**The Lottery Ticket Hypothesis** The lottery ticket hypothesis (LTH) (Frankle & Carbin, 2019) claims the existence of sparse, separate trainable sub-networks in a dense network. These subnetworks are capable of reaching comparable or even better performance than full dense model, which has been evidenced in various fields, such as image classification (Frankle & Carbin, 2019; Liu et al., 2019; Wang et al., 2020a; Evci et al., 2019; Frankle et al., 2020; Savarese et al., 2020; Yin et al., 2020; You et al., 2020; Ma et al., 2021; Chen et al., 2020a), natural language processing (Gale et al., 2019; Chen et al., 2020b), reinforcement learning (Yu et al., 2020), lifelong learning (Chen et al., 2021b), graph neural networks (Chen et al., 2021a), and adversarial robustness (Cosentino et al., 2019). Most works of LTH use unstructured weight magnitude pruning (Han et al., 2016; Frankle &

Carbin, 2019) to find the matching subnetworks, and the channel pruning is also adopted in a recent work (You et al., 2020). In order to scale up LTH to larger networks and datasets, the "late rewinding" technique is proposed by Frankle et al. (2019); Renda et al. (2020). Mehta (2019); Morcos et al. (2019); Desai et al. (2019) are the pioneers to study the transferability of found subnetworks. However, all previous works focus on discriminative models. In this paper, we extend LTH to GANs and reveal unique findings of GAN tickets.

## 3 PRELIMINARIES

In this section, we describe our pruning algorithms and list related experimental settings.

**Backbone Networks**  We use two GANs in our experiments in Section 4: SNGAN (Miyato et al., 2018) and CycleGAN (Zhu et al., 2017)). SNGAN with ResNet (He et al., 2016) is one of the most popular noise-to-image GAN network and has strong performance on several datasets like CIFAR-10. CycleGAN is a popular and well-studied image-to-image GAN network that also performs well on several benchmarks. For SNGAN, let $g(\mathbf{z}; \boldsymbol{\theta_g})$ be the output of the generator network $\mathcal{G}$ with parameters $\boldsymbol{\theta_g}$ and a latent variable $\mathbf{z} \in \mathbb{R}^{\|z\|_0}$ and $d(\mathbf{x}; \boldsymbol{\theta_d})$ be the output of the discriminator network $\mathcal{D}$ with parameters $\boldsymbol{\theta_d}$ and input example $\mathbf{x}$. For CycleGAN which is composed of two generator-discriminator pairs, we use $g(\mathbf{x}; \boldsymbol{\theta_g})$ and $\boldsymbol{\theta_g}$ again to represent the output and the weights of the two generators where $\mathbf{x} = (\mathbf{x}_1, \mathbf{x}_2)$ indicates a pair of input examples. The same modification can be done for the two discriminators in CycleGAN.

**Datasets**  For image-to-image experiments, we use a widely-used benchmark horse2zebra (Zhu et al., 2017) for model training. As for noise-to-image experiments, we use CIFAR-10 (Krizhevsky et al., 2009) as the benchmark. For the transfer study, the experiments are conducted on CIFAR-10 and STL-10 (Coates et al., 2011). For better transferring, we resize the image in STL-10 to $32 \times 32$.

**Subnetworks**  For a network $f(\cdot; \boldsymbol{\theta})$ parameterized by $\boldsymbol{\theta}$, a subnetwork is defined as $f(\cdot; \boldsymbol{m} \odot \boldsymbol{\theta})$, where $\boldsymbol{m} \in \{0, 1\}^{\|\theta\|_0}$ is a pruning mask for $\boldsymbol{\theta} \in \mathbb{R}^{\|\theta\|_0}$ and $\odot$ is the element-wise product. For GANs, two separate masks, $\boldsymbol{m_d}$ and $\boldsymbol{m_g}$, are needed for both the generator and the discriminator. Consequently, a subnetwork of GANs is consistent of: a sparse generator $g(\cdot; \boldsymbol{m_g} \odot \boldsymbol{\theta_g})$ and a sparse discriminator $d(\cdot; \boldsymbol{m_d} \odot \boldsymbol{\theta_d})$.

Let $\boldsymbol{\theta_0}$ be the initialization weights of model $f$ and $\boldsymbol{\theta_t}$ be the weights at training step $t$. Following Frankle et al. (2019), we define a *matching network* as a subnetwork $f(\cdot; \boldsymbol{m} \odot \boldsymbol{\theta})$, where $\boldsymbol{\theta}$ is initialized with $\boldsymbol{\theta_t}$, that can reach the comparable performance to the full network within a similar training iterations when trained in isolation; a *winning ticket* is defined as a matching subnetwork where $t = 0$, *i.e.* $\boldsymbol{\theta}$ initialized with $\boldsymbol{\theta_0}$.

**Finding subnetworks**  Finding GAN subnetworks is to find two masks $\boldsymbol{m_g}$ and $\boldsymbol{m_d}$ for the generator and the discriminator. We use both an unstructured magnitude method, *i.e.* the iterative magnitude pruning (IMP), and a structured pruning method, *i.e.* the channel pruning (He et al., 2017), to generate the masks.

For unstructured pruning, we follow the following steps. After we finish training the full GAN model for $N$ iterations, we prune the weights with the lowest magnitude globally (Han et al., 2016) to obtain masks $\boldsymbol{m} = (\boldsymbol{m_g}, \boldsymbol{m_d})$, where the position of a remaining weight in $\boldsymbol{m}$ is marked as one, and the position of a pruned weight is marked as zero. The weights of the sparse generator and the sparse discriminator are then reset to the initial weights of the full network. Previous works have shown that the iterative magnitude pruning (IMP) method is better than the one-shot pruning method. So rather than pruning the network only once to reach the desired sparsity, we prune a certain amount of non-zero parameters and re-train the network several times to meet the requirement. Details of this algorithm are in Appendix A1.1, Algorithm 1.

As for channel pruning, the first step is to train the full model as well. Besides using a normal loss function $\mathcal{L}_{\text{GAN}}$, we follow Liu et al. (2017) to apply a $\ell_1$-norm on the trainable scale parameters $\gamma$ in the normalization layers to encourage channel-level sparsity: $\mathcal{L}_{\text{cp}} = ||\gamma||_1$. To prevent the compressed network behave severely differently with the original large network, we introduce a distillation loss as Wang et al. (2020b) did: $\mathcal{L}_{\text{dist}} = \mathbb{E}_{\mathbf{z}}[\text{dist}(g(\mathbf{z}; \boldsymbol{\theta_g}), g(\mathbf{z}; \boldsymbol{m_g} \odot \boldsymbol{\theta_g}))]$. We train the

GAN network with these two additional losses for $N_1$ epochs and get the sparse networks $g(\cdot; \boldsymbol{m_g} \odot \boldsymbol{\theta_g})$ and $d(\cdot; \boldsymbol{m_d} \odot \boldsymbol{\theta_g})$. Details of this algorithm are in Appendix A1.1, Algorithm 2.

**Evaluation of subnetworks**    After obtaining the subnetworks $g(\cdot; \boldsymbol{\theta_g} \odot \boldsymbol{m_g})$ and $d(\cdot; \boldsymbol{\theta_d} \odot \boldsymbol{m_d})$, we test whether the subnetworks are matching or not. We reset the weights to a specific step $i$, and train the subnetworks for $N$ iterations and evaluate them using two specific metrics, Inception Score (Salimans et al., 2016) and Fréchet Inception Distance (Heusel et al., 2017).

**Other Pruning Methods**    We compare the size and the performance of subnetworks found by IMP with subnetworks found by other techniques that aim at compressing the network after training to reduce computational costs at inference. We use a benchmark pruning approach named *Standard Pruning* (Chen et al., 2020b; Han et al., 2016), which iteratively prune the 20% of lowest magnitude weights, and train the network for another $N$ iterations without any rewinding, and repeat until we have reached the target sparsity.

In order to verify that the statement of iterative magnitude pruning is better than one-shot pruning, we compare $\text{IMP}_\text{G}$ and $\text{IMP}_\text{GD}$ with their one-shot counterparts. Additionally, we compare IMP with some randomly pruning techniques to prove the effectiveness of IMP. They are: 1) *Randomly Pruning*: Randomly generate a sparsity mask $\boldsymbol{m}'$. 2) *Random Tickets*: Rewinding the weights to another initialization $\boldsymbol{\theta_0'}$.

# 4    THE EXISTENCE OF WINNING TICKETS IN GAN

In this section, we will validate the existence of winning tickets in GANs with initialization $\boldsymbol{\theta_0} := (\boldsymbol{\theta_{g_0}}, \boldsymbol{\theta_{d_0}})$. Specifically, we will empirically prove several important properties of the tickets by authenticating the following four claims:

*Claim 1*: Iterative Magnitude Pruning (IMP) finds winning tickets in GANs, $g(\cdot; \boldsymbol{m_g} \odot \boldsymbol{\theta_{g_0}})$ and $d(\cdot; \boldsymbol{m_d} \odot \boldsymbol{\theta_{d_0}})$. Channel pruning is also able to find winning tickets as well.

*Claim 2*: Whether pruning the discriminator $\mathcal{D}$ does not change the existence of winning tickets. It is the initialization used in $\mathcal{D}$ that matters. Moreover, pruning the discriminator has a slight boost of matching networks in terms of extreme sparsity and performance.

*Claim 3*: IMP finds winning tickets at sparsity where some other pruning methods (randomly pruning, one-shot magnitude pruning, and random tickets) are not matching.

*Claim 4*: The late rewinding technique (Frankle et al., 2019) helps. Matching subnetworks that are initialized to $\boldsymbol{\theta_i}$, *i.e.*, $i$ steps from $\boldsymbol{\theta_0}$, can outperform those initialized to $\boldsymbol{\theta_0}$. Moreover, matching subnetworks that are late rewound can be trained to match the performance of standard pruning.

**Claim 1: Are there winning tickets in GANs?**    To answer this question, we first conduct experiments on SNGAN by pruning the generator only in the following steps: 1) Run IMP to get sequential sparsity masks $(\boldsymbol{m_{d_i}}, \boldsymbol{m_{g_i}})$ of sparsity $s_i\%$ remaining weights; 2) Apply the masks to the GAN and reset the weights of the subnetworks to the same random initialization $\boldsymbol{\theta_0}$; 3) Train models to evaluate whether they are winning tickets. [1] We set $s_i\% = (1 - 0.8^i) \times 100\%$, which we use for all the experiments that involve iteratively pruning hereafter. The number of training epochs for subnetworks is identical to that of training the full models.

Figure 1 verifies the existence of winning tickets in SNGAN and CycleGAN. We are able to find winning tickets by iterative pruning the generators at the highest sparsity, around 74% in SNGAN, and around 67% in CycleGAN, where the FID scores of these subnetworks successfully match the FID scores of the full network respectively. The confidence interval also suggests that the winning tickets at some sparsities are statistically significantly better than the full model.

To show that channel pruning can find winning tickets as well, we extract several subnetworks from the trained full SNGAN and CycleGAN by varying $\rho$ in Algorithm 2. We define the channel-pruned model's sparsity as the ratio of MFLOPs between the sparse model and the full model.

---

[1]The detailed description of algorithms are listed in Appendix A1.1.

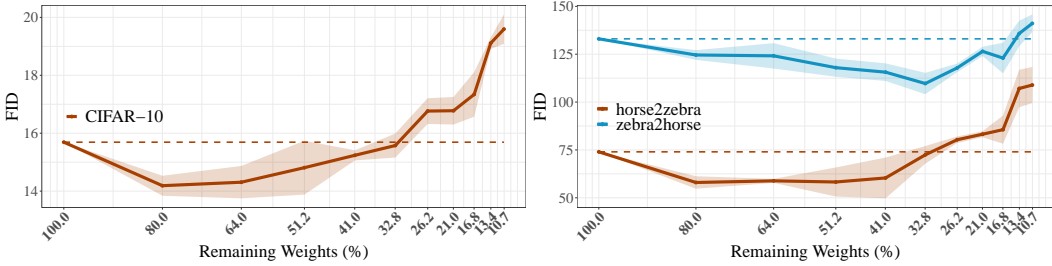

Figure 1: The Fréchet Inception Distance (FID) curve of subnetworks of SNGAN (left) and CycleGAN (right) generated by iterative magnitude pruning (IMP) on CIFAR-10 and horse2zebra. The dashed line indicates the FID score of the full model on CIFAR-10 and horse2zebra. The 95% confidence interval of 5 runs is reported.

We can confirm that winning tickets can also be found by channel pruning (CP). CP is able to find winning tickets in SNGAN at sparsity around 34%. We will analyze it more carefully in Section 7.

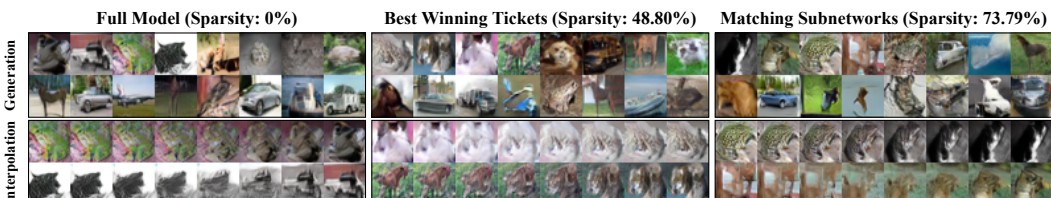

Figure 2: Visualization by sampling and interpolation of SNGAN Winning Tickets found by IMP. Sparsity of best winning tickets : 48.80%. Extreme sparsity of matching subnetworks: 73.79%.

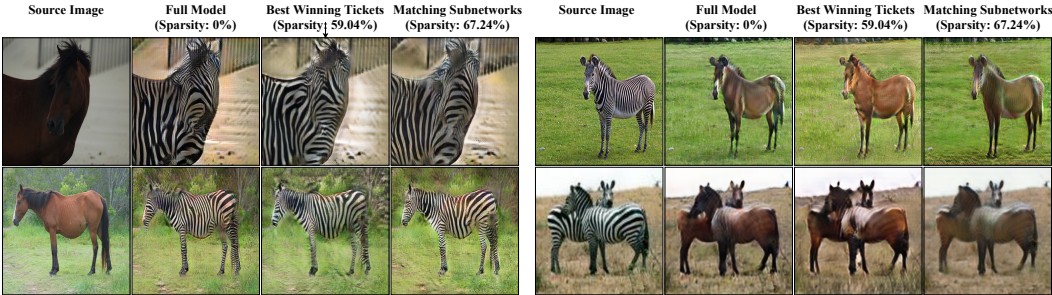

Figure 3: Visualization of CycleGAN Winning Tickets found by IMP. Sparsity of best winning tickets : 59.04%. Extreme sparsity of matching subnetworks: 67.24%. Left: visualization results on horse2zebra. Right: visualization results on zebra2horse.

**Claim 2: Does the treatment of the discriminator affect the existence of winning tickets?** Previous works of GAN pruning did not analyze the effect of pruning the discriminator. To study the effect, we compare two different iterative pruning settings: 1) Prune the generator only ($\mathrm{IMP_G}$) and 2) Prune both the generator and the discriminator iteratively ($\mathrm{IMP_{GD}}$). Both the generator and the discriminator are reset to the same random initialization $\theta_0$ after the masks are obtained.

The FID scores of the two experiments are shown in Figure 4. The graph suggests that the two settings share similar patterns: the minimal FID of $\mathrm{IMP_G}$ is 14.19, and the minimal FID of $\mathrm{IMP_{GD}}$ is 14.59. The difference between these two best FID is only 0.4, showing a slight difference in generative power. The FID curve of $\mathrm{IMP_G}$ lies below that of $\mathrm{IMP_{GD}}$ at low sparsity but lies above at high sparsity, indicating that pruning the discriminator produces slightly better performance when the percent of remaining weights is small. The extreme sparsity where $\mathrm{IMP_{GD}}$ can match the performance of the full model is 73.8%. In contrast, $\mathrm{IMP_G}$ can only match no sparser than 67.2%, demonstrating that pruning the discriminator can also push the frontier of extreme sparsity where the pruned models are still able to match.

In addition, we study the effects of different initialization for the sparse discriminator. We compare different weights loading methods when applying the iterative magnitude pruning process: 1) Reset the weights of generator to $\theta_{g_0}$ and reset the weights of discriminator to $\theta_{d_0}$, which is identical to $\mathrm{IMP_G}$; 2) Reset the weights of generator to $\theta_{g_0}$ and **fine-tune** the discriminator, which we will call $\mathrm{IMP_G^F}$. Figure 4 shows that resetting both the weights to $\theta_0$ produces a much better result than

only resetting the generator. The discriminator $\mathcal{D}$ without resetting its weights is too strong for the generator $\mathcal{G}$ with initial weights $\boldsymbol{\theta_{g_0}}$ that will lead to degraded performance. In summary, different initialization of the discriminator will significantly influence the existence and quality of winning tickets in GAN models.

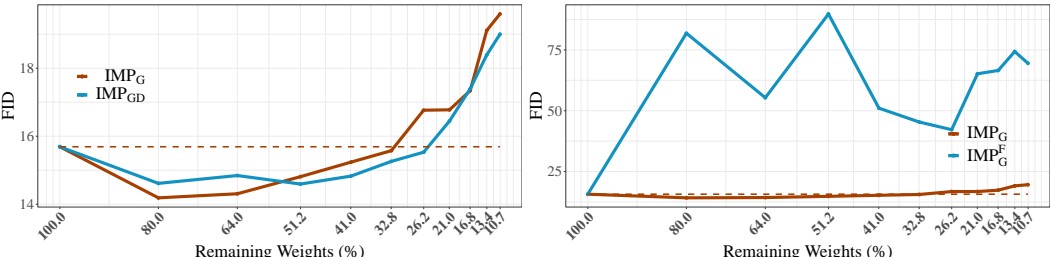

Figure 4: The FID score of Left: The FID score of best subnetworks generated by two different pruning settings: $\text{IMP}_\text{G}$ and $\text{IMP}_\text{GD}$. Right: The FID score of best subnetworks generated by two different pruning settings: $\text{IMP}_\text{G}$ and $\text{IMP}_\text{G}^\text{F}$. $\text{IMP}_\text{G}$: iteratively prune and reset the generator. $\text{IMP}_\text{GD}$: iteratively prune and reset the generator and the discriminator. $\text{IMP}_\text{G}^\text{F}$: iteratively prune and reset the generator, and iteratively prune but not reset the discriminator.

A follow-up question arises from the previous observations: is there any way to use the weights of the dense discriminator, as the direct usage of the dense weights yields inferior results? One possible way is to use it as a "teacher" and transfer the knowledge to pruned discriminator using a consistency loss. Formally speaking, an additional regularization term is used when training the whole network:

$$\mathcal{L}_{\text{KD}}(\mathbf{x}; \boldsymbol{\theta_d}, \boldsymbol{m_d}) = \mathbb{E}_\mathbf{x}[\text{KL}_{\text{Div}}(d(\mathbf{x}; \boldsymbol{m_d} \odot \boldsymbol{\theta_d}), d(\mathbf{x}; \boldsymbol{\theta_{d_1}}))]$$

where $\text{KL}_{\text{Div}}$ denotes the KL-Divergence. We name the iterative pruning method with this additional regularization $\text{IMP}_\text{GD}^\text{KD}$.

Figure 5 shows the result of pruning method $\text{IMP}_\text{GD}^\text{KD}$ compared to the previous two pruning methods we proposed, $\text{IMP}_\text{G}$ and $\text{IMP}_\text{GD}$. $\text{IMP}_\text{GD}^\text{KD}$ is capable of finding winning tickets at sparsity around 70%, outperforming $\text{IMP}_\text{G}$, and showing comparable results to $\text{IMP}_\text{GD}$ regarding the extreme sparsity. The FID curve of setting $\text{IMP}_\text{GD}^\text{KD}$ is further mostly located below the curve of $\text{IMP}_\text{GD}$, demonstrating a stronger generative ability than $\text{IMP}_\text{GD}$. It suggests transferring knowledge from the full discriminator benefits to find the winning tickets.

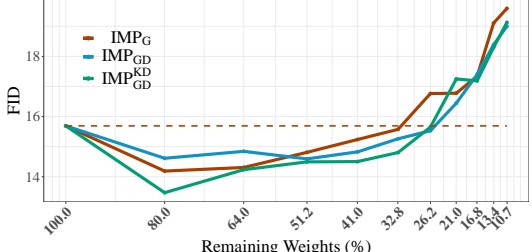

Figure 5: The FID curve of best subnetworks generated by three different pruning methods: $\text{IMP}_\text{G}$, $\text{IMP}_\text{GD}$ and $\text{IMP}_\text{GD}^\text{KD}$. $\text{IMP}_\text{GD}^\text{KD}$: iteratively prune and reset both the generator and the discriminator, and train them with the KD regularization.

**Claim 3: Can IMP find matching subnetworks sparser than other pruning methods?** Previous works claim that both a specific sparsity mask and a specific initialization are necessary for finding winning tickets (Frankle & Carbin, 2019), and iterative magnitude pruning is better than one-shot pruning. To extend such a statement in the context of GANs, we compare IMP with several other benchmarks, randomly pruning (RP), one-shot magnitude pruning (OMP), and random tickets (RT), to see if IMP can find matching networks at higher sparsity.

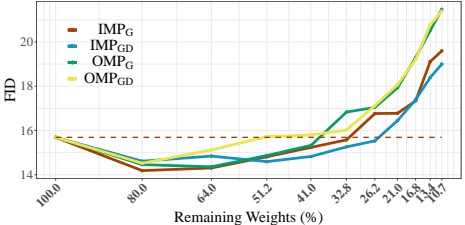

Figure 6: FID curves: $\text{OMP}_\text{G}$, $\text{IMP}_\text{G}$, $\text{OMP}_\text{GD}$ and $\text{IMP}_\text{GD}$. $\text{OMP}_\text{G}$: one-shot prune the generator. $\text{OMP}_\text{GP}$: one-shot prune generator/discriminator.

| Method | $\text{FID}_\text{Best}$ (Sparsity) | $\text{FID}_\text{Extreme}$ (Sparsity) |
|---|---|---|
| No Pruning | 15.69 (0.0%) | - |
| $\text{IMP}_\text{G}$ | 14.19 (20.0%) | 15.58 (67.2%) |
| $\text{IMP}_\text{GD}$ | 14.59 (59.0%) | 15.53 (73.8%) |
| $\text{OMP}_\text{G}$ | 14.36 (36.0%) | 15.33 (59.0%) |
| $\text{OMP}_\text{GD}$ | 14.52 (20.0%) | 15.69 (48.8%) |

Table 1: The extreme sparsity of the matching networks and the FID score of best subnetworks, found by iterative pruning and one-shot pruning. $\text{OMP}_\text{G}$: one-shot prune the generator. $\text{OMP}_\text{GP}$: one-shot prune generator/discriminator.

Figure 6 and Table 1 show that iterative magnitude pruning outperforms one-shot magnitude pruning no matter pruning the discriminator or not. IMP finds winning tickets at higher sparsity (67.23% and 73.79%, respectively) than one-shot pruning (59.00% and 48.80%, respectively). The minimal FID score of subnetworks found by IMP is smaller than that of subnetworks found by OMP as well. This observation defends the statement that pruning iteratively is superior compared to one-shot pruning.

We also list the minimal FID scores and extreme sparsity of matching networks for other pruning methods in Figure 7 and Table 2. It can be seen that IMP finds winning tickets at sparsity where some other pruning methods, randomly pruning and random initialization, cannot match. Since $\text{IMP}_G$ shows the best overall result, we authenticate the previous statement that both the specific sparsity mask and the specific initialization are essential for finding winning tickets.

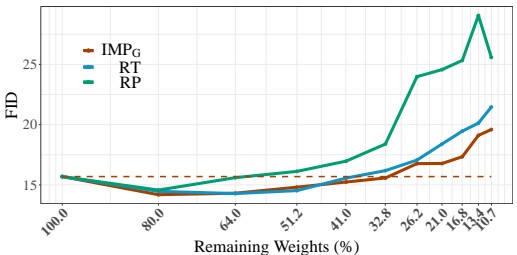

Figure 7: The FID curve of best subnetworks generated by three different pruning settings: $\text{IMP}_G$, RP, and RT. RP: iteratively *randomly* prune the generator. RT: iteratively prune the generator but reset the weights *randomly*.

| Method | $\text{FID}_{\text{Best}}$ (Sparsity) | $\text{FID}_{\text{Extreme}}$ (Sparsity) |
|---|---|---|
| No Pruning | 15.69 (0.0%) | |
| $\text{IMP}_G$ | 14.19 (20.0%) | 15.58 (67.2%) |
| Random Pruning | 14.57 (20.0%) | 15.60 (36.0%) |
| Random Tickets | 14.28 (36.0%) | 15.33 (59.0%) |

Table 2: The FID score of best subnetworks and the extreme sparsity of matching networks found by Random Pruning, Random Rickets, and iterative magnitude pruning.

**Claim 4: Does rewinding improve performance?** In previous paragraphs, we show that we are able to find winning tickets in both SNGAN and CycleGAN. However, these subnetworks cannot match the performance of the original network at extremely high sparsity, while the subnetworks found by standard pruning can (Table 3). To find matching subnetworks at such high sparsity, we adopt the *rewinding* paradigm: after the masks are obtained, the weights of the model are rewound to $\theta_i$, the weights after $i$ steps of training, rather than reset to the same random initialization $\theta_0$. It was pointed out by Renda et al. (2020) that subnetworks found by IMP and rewound early in training can be trained to achieve the same accuracy at the same sparsity as subnetworks found by the standard pruning, providing a possibility that rewinding can also help GAN subnetworks.

We choose different rewinding settings: 5%, 10%, and 20% of the whole training epochs. The results are shown in Table 3. We observe that rewinding can significantly increase the extreme sparsity of matching networks. Rewinding to even only 5% of the training process can raise the extreme sparsity from 67.23% to 86.26%, and rewinding to 20% can match the performance of standard pruning. We also compare the FID score of subnetworks found at 89% sparsity. Rewind to 20% of the training

Table 3: Rewinding results. $\text{S}_{\text{Extreme}}$: Extreme sparsity where matching subnetworks exist. $\text{FID}_{\text{Best}}$: The minimal FID score of all subnetworks. $\text{FID}_{89\%}$: The FID score of subnetworks at 89% sparsity.

| Setting of rewinding | $\text{S}_{\text{Extreme}}$ | $\text{FID}_{\text{Best}}$ | $\text{FID}_{89\%}$ |
|---|---|---|---|
| Rewind 0% | 67.23% | 14.20 | 19.60 |
| Rewind 5% | 86.26% | **13.96** | 15.82 |
| Rewind 10% | 86.26% | 14.43 | 15.63 |
| Rewind 20% | **89.26%** | 14.82 | 15.29 |
| Standard Pruning | **89.26%** | 14.18 | **15.22** |

process can match the performance of standard pruning at 89% sparsity, and other late rewinding settings can match the performance of the full model. This suggests that late rewinding techniques can greatly contribute to matching subnetworks with higher sparsity.

**Summary** Extensive experiments are conducted to examine the existence of matching subnetworks in generative adversarial models. We confirmed that there were matching subnetworks at high sparsities, and both the sparsity mask and the initialization matter for finding winning tickets. We also studied the effect of pruning the discriminator and demonstrate that pruning the discriminator can slightly boost the performance regarding the extreme sparsity and the minimal FID. We proposed a method to utilize the weights of the dense discriminator model to boost the performance further. We also compare IMP with different pruning methods, showing that IMP is superior to random tickets and random pruning. In addition, late rewinding can match the performance of standard pruning, which again shows consistency with previous works.

## 5 THE TRANSFER LEARNING OF GAN MATCHING NETWORKS

In the previous section, we confirm the presence of winning tickets in GANs. In this section, we will study the transferability of winning tickets. Existing works (Mehta, 2019) show that the matching networks in discriminative models can transfer across tasks. Here we evaluate this claim in GANs.

To investigate the transferability, we propose three transfer experiments from CIFAR-10 to STL-10 on SNGAN. We first identify matching subnetworks $g(\cdot, \boldsymbol{m_g} \odot \boldsymbol{\theta})$ and $d(\cdot, \boldsymbol{m_d} \odot \boldsymbol{\theta})$ on CIFAR-10, and then train and evaluate the subnetworks on STL-10. To assess whether the same random initialization $\boldsymbol{\theta_0}$ is needed for transferring, we test three different weights loading method: 1) reset the weights to $\boldsymbol{\theta_0}$; 2) reset the weights to another initialization $\boldsymbol{\theta_0'}$; 3) rewind the weights to $\boldsymbol{\theta_N}$. We train the network on STL-10 using the same hyper-parameters as on CIFAR-10. It is noteworthy that the hyper-parameters setting might not be optimal for the target task, yet it is fair to compare different transferring settings.

Table 4: Results of late rewinding experiments. $\theta_0$: train the target model from the same random initialization as the source model; $\boldsymbol{\theta_r}$: train from random initialization; $\boldsymbol{\theta_{Best}}$: train from the weights of trained source model. Baseline: full model trained on STL-10.

| Model | Baseline | $\mathrm{IMP_G}$ (S = 67.23%) | | $\mathrm{IMP_{GD}}$ (S = 73.79%) | | $\mathrm{IMP_{GD}^{KD}}$ (S = 73.79%) | |
|---|---|---|---|---|---|---|---|
| Metrics | $\mathrm{FID_{Best}}$ | $\mathrm{FID_{Best}}$ | Matching? | $\mathrm{FID_{Best}}$ | Matching? | $\mathrm{FID_{Best}}$ | Matching? |
| $\boldsymbol{\theta_0}$ | | **116.7** | ✓ | 121.8 | ✗ | 120.1 | ✗ |
| $\boldsymbol{\theta_r}$ | 115.3 | 119.2 | ✗ | **113.1** | ✓ | **115.5** | ✓ |
| $\boldsymbol{\theta_{Best}}$ | | 204.7 | ✗ | 163.0 | ✗ | 179.1 | ✗ |

The FID score of different settings is shown in Table 4. Subnetworks initialized by $\theta_0$ and using masks generated by $\mathrm{IMP_G}$ can be trained to achieve comparable results to the baseline model. Surprisingly, random re-initialization $\theta_0'$ shows better transferability than using the same initialization $\boldsymbol{\theta_0}$ in our transfer settings and outperforms the full model trained on STL-10, indicating that the combination of $\boldsymbol{\theta_0}$ and the mask generated by $\mathrm{IMP_{GD}}$ is more focused on the source dataset and consequently has lower transferability.

**Summary**  In this section, we tested the transferability of IMP subnetworks. Transferring from $\boldsymbol{\theta_0}$ and $\boldsymbol{\theta_r}$ both produce matching results on the target dataset, STL-10. $\boldsymbol{\theta_0}$ works better with masks generated by $\mathrm{IMP_G}$ while the masks generated by $\mathrm{IMP_{GD}}$ prefer a different initialization $\boldsymbol{\theta_r}$. Given that $\mathrm{IMP_{GD}}$ performs better on CIFAR-10, it is reasonable that the same initialization $\boldsymbol{\theta_0}$ has lower transferability when using masks from $\mathrm{IMP_{GD}}$.

## 6 EXPERIMENTS ON OTHER GAN MODELS AND OTHER DATASETS

We conducted experiments on DCGAN (Radford et al., 2016), WGAN-GP (Gulrajani et al., 2017), ACGAN (Odena et al., 2017), GGAN (Lim & Ye, 2017), DiffAugGAN (Zhao et al., 2020a), ProjGAN (Miyato & Koyama, 2018), SAGAN (Zhang et al., 2019), as well as a NAS-based GAN, AutoGAN (Gong et al., 2019). We use CIFAR-10 and Tiny ImageNet (Wu et al., 2017) as our benchmark datasets.

Table 5 and 6 consistently verify that the existence of winning tickets in diverse GAN architectures in spite of the different extreme sparsities, showing that the lottery ticket hypothesis can be generalized to various GAN models.

## 7 EFFICIENCY OF GAN WINNING TICKETS

Unlike the unstructured magnitude pruning method, channel pruning can reduce the number of parameters in GANs. Therefore, winning tickets found by channel pruning are more efficient than the original model regarding computational cost. To fully exploit the advantage of subnetworks founded by structural pruning, we further compare our prune-and-train pipeline with a state-of-the-art GAN compression framework (Wang et al., 2020b). The pipeline is described as follows: after extracting the sparse structure generated by channel pruning, we reset the model weights to the same random initialization $\boldsymbol{\theta_0}$ and then train for the same number of epochs as the dense model used.

Table 5: Results on other GAN models on CIFAR-10. $FID_{Full}$: FID score of the full model. $FID_{Best}$: The minimal FID score of all subnetworks. $FID_{Extreme}$: The FID score of matching networks at extreme sparsity level. AutoGAN-A/B/C are three representative GAN architectures represented in the official repository (https://github.com/VITA-Group/AutoGAN)

| Model | Benchmark | $FID_{Full}$ (Sparsity) | $FID_{Best}$ (Sparsity) | $S_{Extreme}$ (Sparsity) |
|---|---|---|---|---|
| DCGAN (Radford et al., 2016) | CIFAR-10 | 57.39 (0%) | 49.31 (20.0%) | 54.48 (67.2%) |
| WGAN-GP (Gulrajani et al., 2017) | CIFAR-10 | 19.23 (0%) | 16.77 (36.0%) | 17.28 (73.8%) |
| ACGAN (Odena et al., 2017) | CIFAR-10 | 39.26 (0%) | 31.45 (36.0%) | 38.95 (79.0%) |
| GGAN (Lim & Ye, 2017) | CIFAR-10 | 38.50 (0%) | 33.42 (20.0%) | 36.67 (48.8%) |
| ProjGAN (Miyato & Koyama, 2018) | CIFAR-10 | 31.47 (0%) | 28.19 (20.0%) | 31.31 (67.2%) |
| SAGAN (Zhang et al., 2019) | CIFAR-10 | 14.73 (0%) | 13.57 (20.0%) | 14.68 (48.8%) |
| AutoGAN(A) (Gong et al., 2019) | CIFAR-10 | 14.38 (0%) | 14.04 (36.0%) | 14.04 (36.0%) |
| AutoGAN(B) (Gong et al., 2019) | CIFAR-10 | 14.62 (0%) | 13.16 (20.0%) | 14.20 (36.0%) |
| AutoGAN(C) (Gong et al., 2019) | CIFAR-10 | 13.61 (0%) | 13.41 (48.8%) | 13.41 (48.8%) |
| DiffAugGAN (Zhao et al., 2020b) | CIFAR-10 | 8.23 (0%) | 8.05 (48.8%) | 8.05 (48.8%) |

Table 6: Results on other GAN models on Tiny ImageNet. $FID_{Full}$: FID score of the full model. $FID_{Best}$: The minimal FID score of all subnetworks. $FID_{Extreme}$: The FID score of matching networks at extreme sparsity level.

| Model | Benchmark | $FID_{Full}$ (Sparsity) | $FID_{Best}$ (Sparsity) | $S_{Extreme}$ (Sparsity) |
|---|---|---|---|---|
| DCGAN (Radford et al., 2016) | Tiny ImageNet | 121.35 (0%) | 78.51 (36.0%) | 114.00 (67.2%) |
| WGAN-GP (Gulrajani et al., 2017) | Tiny ImageNet | 211.77 (0%) | 194.72 (48.8%) | 200.22 (67.2%) |

We can see from Figure 8 that subnetworks are founded by the channel pruning method at about 67% sparsity, which provides a new path for winning tickets other than magnitude pruning. The matching networks at 67.7% outperform GS-32 regarding Inception Score by 0.25; the subnetworks at about 29% sparsity can outperform GS-32 by 0.20, setting up a new benchmark for GAN compressions.

# 8    CONCLUSION

In this paper, the lottery ticket hypothesis has been extended to GANs. We successfully identify winning tickets in GANs, which are separately trainable to match the full dense GAN performance. Pruning the discriminator, which is rarely studied before, had only slight effects on the ticket finding process, while the initialization used in the discriminator is essential. We also demonstrate that the winning tickets found can transfer across diverse tasks. Moreover, we provide a new way of finding winning tickets that alter the structure of models. Channel pruning is able to extract matching subnetworks from a dense model that can outperform the current state-of-the-art GAN compression after resetting the weights and re-training.

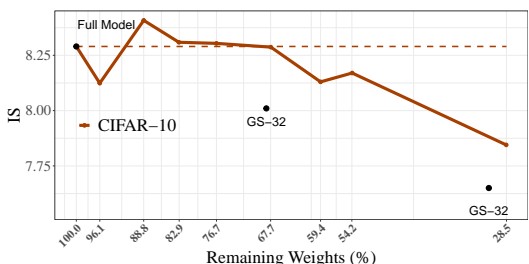

Figure 8: Relationship between the best IS score of SNGAN subnetworks generated by channel pruning and the percent of remaining weights. GS-32: GAN Slimming without quantization (Wang et al., 2020b). Full Model: Full model trained on CIFAR-10.

## ACKNOWLEDGEMENT

Zhenyu Zhang is supported by the National Natural Science Foundation of China under grand No.U19B2044.

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

## A1   MORE TECHNICAL DETAILS

### A1.1   ALGORITHMS

In this section, we describe the details of the algorithm we used in finding lottery tickets. Two distinct pruning methods are used in Algorithm 1 and Algorithm 2.

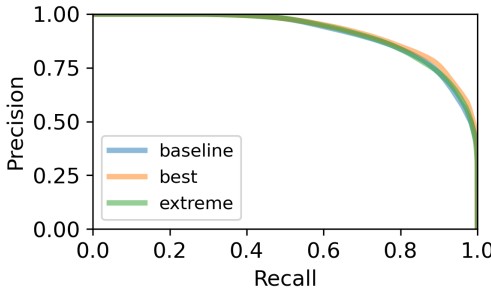

| Model | $F_8$ | $F_{1/8}$ |
|---|---|---|
| Full Model | 0.971 | 0.974 |
| Best Winning Tickets | **0.977** | **0.977** |
| Extreme Winning Tickets | 0.974 | 0.971 |

Table A7: The $F_8$ and $F_{1/8}$ score of the full network, best subnetworks and the matching networks at extreme sparsity. We used the official codes to calculate recall, precision, $F_8$ and $F_{1/8}$.

Figure A9: The curve of precision and recall of SNGANs under different sparsities. *baseline*: Full model. *best*: Best winning tickets (Sparsity: 48.80%). *extreme*: Extreme winning tickets (Sparsity: 73.79%).

## A2   MORE EXPERIMENTS RESULTS AND ANALYSIS

We will provide extra experiments results and analysis in this section.

### A2.1   MORE VISUALIZATION OF IMP WINNING TICKETS

We also conducted experiments to find winning tickets in CycleGAN on dataset winter2summer (Zhu et al., 2017). We observed similar patterns and found matching networks at 79.02% sparsity. We randomly sample four images from the dataset and show the translated images in Figure A10, Figure A11, and Figure A12. The winning tickets of CycleGAN can generate comparable visual quality to the full model under all cases.

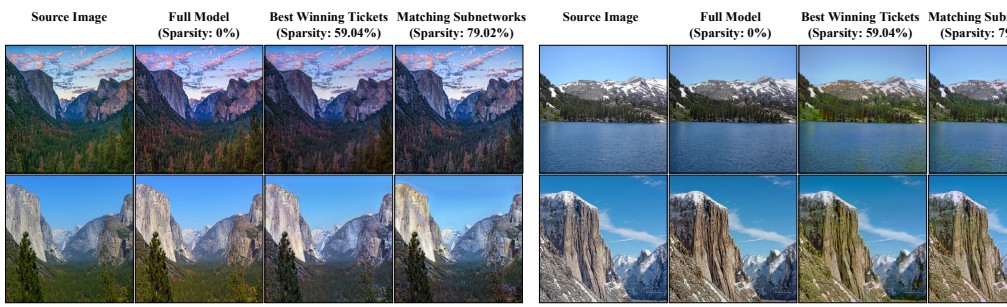

Figure A10: Visualization of CycleGAN Winning Tickets found by IMP on summer2winter. Sparsity of best winning tickets: 59.04%. Extreme sparsity of matching subnetworks: 79.02%. Left: visualization results of task summer2winter. Right: visualization results of task winter2summer.

### A2.2   EXTRA METRICS FOR EVALUATING SNGAN

We also evaluate the quality of images generated by SNGAN using precision and recall Sajjadi et al. (2018). The results are shown in Table A7 and Figure A9. The results show that the best winning tickets have higher $F_8$ and $F_{1/8}$ compared to the full model, and the extreme winning tickets have on-par performance.

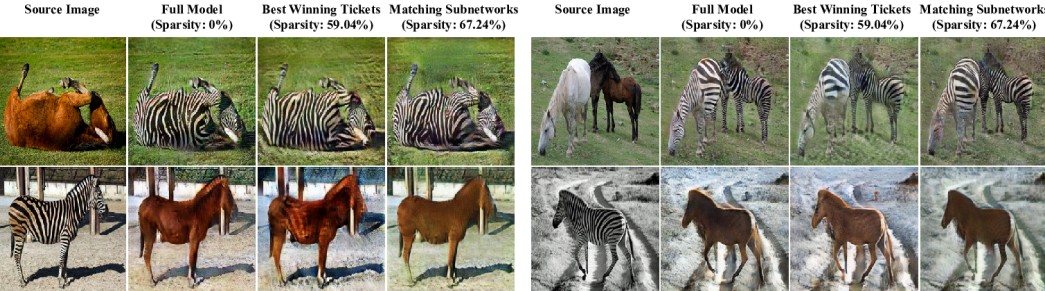

Figure A11: Extra visualization of CycleGAN Winning Tickets found by IMP on horse2zebra. Sparsity of best winning tickets : 59.04%. Extreme sparsity of matching subnetworks: 67.24%.

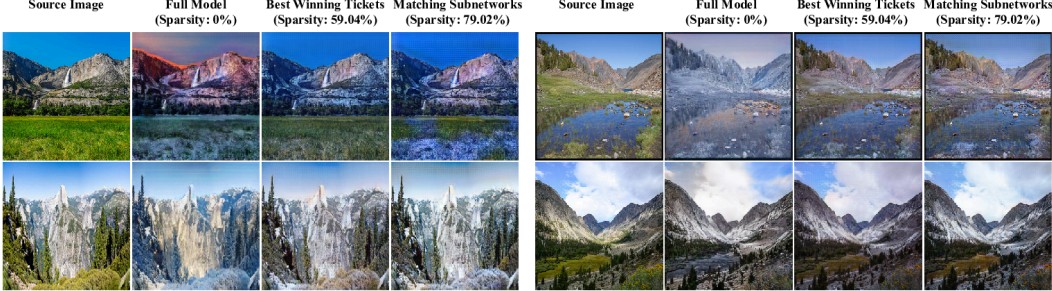

Figure A12: Extra visualization of CycleGAN Winning Tickets found by IMP on summer2winter. Sparsity of best winning tickets : 59.04%. Extreme sparsity of matching subnetworks: 79.02%.

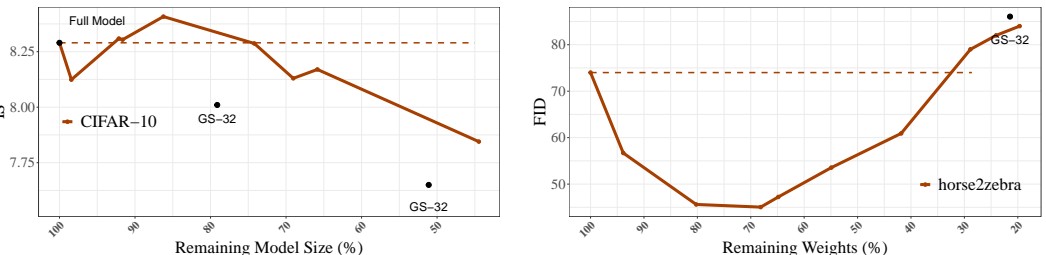

Figure A13: Relationship between the best IS score of SNGAN subnetworks generated by channel pruning and the percent of remaining model size. GS-32: GAN Slimming without quantization (Wang et al., 2020b). Full Model: Full model trained on CIFAR-10.

Figure A14: Relationship between the best FID score of CycleGAN subnetworks generated by channel pruning and the percent of remaining weights. GS-32: GAN Slimming without quantization (Wang et al., 2020b). Full Model: Full un-pruned CycleGAN trained on horse2zebra.

### A2.3 CHANNEL PRUNING FOR SNGAN

We study the relationship between the Inception Score and the *remaining model size*, *i.e.* the ratio between the size of a channel-pruned model and its original model. The results are plotted in Figure A13. A similar conclusion can be drawn from the graph that matching networks exist, and at the same sparsity, the matching networks can be trained to outperform the current state-of-the-art GAN compression framework.

### A2.4 CHANNEL PRUNING FOR CYCLEGAN

We also conducted experiments on CycleGAN using channel pruning. The task we choose is horse-to-zebra, *i.e.*, we prune each of the two generators separately, which is aligned with SNGAN, which has only one generator. We prove that channel pruning is also capable of finding winning tickets in CycleGAN in Figure A14. Moreover, at extreme sparsity, the sparse subnetwork that we obtain

can be trained to reach slightly better results than the current state-of-the-art GAN compression framework without quantization.

---

**Algorithm 1:** Finding winning tickets by Iterative Magnitude Pruning

---

**Input:** The desired sparsity $s$
**Output:** A sparse GAN $g(\cdot; \boldsymbol{m_g} \odot \boldsymbol{\theta_g})$
and $d(\cdot; \boldsymbol{m_d} \odot \boldsymbol{\theta_d})$

1 Set $\boldsymbol{m_g} = \mathbf{1} \in \mathbb{R}^{\|\boldsymbol{\theta_{g0}}\|_0}$ and
$\boldsymbol{m_d} = \mathbf{1} \in \mathbb{R}^{\|\boldsymbol{\theta_{d0}}\|_0}$.
2 Set $\boldsymbol{\theta_{g0}} :=$ initial weights of the generator model, $\boldsymbol{\theta_{d0}} :=$ initial weights of the discriminator model.
3 Iteration $i = 0$
4 **while** *the sparsity of $\boldsymbol{m_g} < s$* **do**
5     Train the generator $g(\cdot; \boldsymbol{m_g} \odot \boldsymbol{\theta_{g0}})$ and the discriminator $d(\cdot; \boldsymbol{m_d} \odot \boldsymbol{\theta_{d0}})$ for $N$ epochs to get parameters $\boldsymbol{\theta^i_{g_N}}$ and $\boldsymbol{\theta^i_{d_N}}$.
6     **if** *pruning the discriminator* **then**
7         Prune 20% of the parameters in $\boldsymbol{\theta^i_{g_N}}$ and $\boldsymbol{\theta^i_{d_N}}$, creating two mask $\boldsymbol{m'_g}$ and $\boldsymbol{m'_d}$.
8     **else**
9         Prune 20% of the parameters in $\boldsymbol{\theta^i_{g_N}}$, creating a mask $\boldsymbol{m'_g}$. $\boldsymbol{m'_d}$ remains $\mathbf{1} \in \mathbb{R}^{\|\boldsymbol{\theta_{d0}}\|_0}$.
10     **end**
11     Update $\boldsymbol{m_g} = \boldsymbol{m'_g}$ and $\boldsymbol{m_d} = \boldsymbol{m'_d}$
12     $i = i + 1$
13 **end**

---

**Algorithm 2:** Finding winning tickets by Channel Pruning

---

**Input:** A threshold $\rho$ of importance score, number of steps for training $N$
**Output:** A sparse GAN $g(\cdot; \boldsymbol{m_g} \odot \boldsymbol{\theta_{g0}})$ and $d(\cdot; \boldsymbol{m_d} \odot \boldsymbol{\theta_{d0}})$

1 Randomly initialize $\boldsymbol{\gamma_g}$ for every normalization layers in generator $\mathcal{G}$ and $\boldsymbol{\gamma_d}$ for discriminator $\mathcal{D}$. $\boldsymbol{\gamma} = (\boldsymbol{\gamma_g}, \boldsymbol{\gamma_d})$
2 Set $\boldsymbol{\theta_{g0}} :=$ initial weights of the generator model, $\boldsymbol{\theta_{d0}} :=$ initial weights of the discriminator model.
3 $i = 0$
4 **while** $i < N$ **do**
5     Compute masks $\boldsymbol{m_d}$ from $\boldsymbol{\gamma_d}$ and $\boldsymbol{m_g}$ from $\boldsymbol{\gamma_g}$.
6     Compute $\mathcal{L}_{\text{cp}}$, $\mathcal{L}_{\text{GAN}}$ and $\mathcal{L}_{\text{dist}}$.
7     Update $\boldsymbol{\theta_g}$ and $\boldsymbol{\theta_d}$ by training the generator $g(\cdot; \boldsymbol{m_g} \odot \boldsymbol{\theta_g})$ and the discriminator $d(\cdot; \boldsymbol{m_d} \odot \boldsymbol{\theta_d})$ for one step.
8     Update $\boldsymbol{\gamma}$: $\boldsymbol{\gamma} \leftarrow \text{prox}_{\rho\eta}(\boldsymbol{\gamma} - \eta\nabla_{\boldsymbol{\gamma}}\mathcal{L}_{\text{cp}})$, where $\text{prox}_\lambda(x) = \text{sgn}(x) \odot \max(|x| - \lambda \cdot \mathbf{1}, 0)$
9     $i \leftarrow i + 1$
10 **end**
11 Calculate the final masks $\boldsymbol{m_g}$ from $\boldsymbol{\gamma_g}$ and $\boldsymbol{m_d}$ from $\boldsymbol{\gamma_d}$

---

