# OpenReview forum: "GANs Can Play Lottery Tickets Too"
_ICLR.cc/2021/Conference — ICLR 2021 Poster_

### Official Review · AnonReviewer2 · 2020-10-26

**Rating:** 8
**Confidence:** 3

**Review:**

**Summary**

The authors study the lottery ticket hypothesis  for generative adversarial networks. Specifically, they attempt to answer the following questions: the existence of winning tickets in GANs; the effect of discriminator pruning in finding such winning tickets; the effect of initialization during the rewinding steps; and finally if the subnetworks found transfer across datasets. They provide extensive empirical evidence using that ```winning' tickets exist in GANs. Further they show that iterative magnitude pruning and channel pruning successfully find such `winning' subnetworks. They analyse the effect of discriminator pruning and find that initialization during the rewind step matters more than the actual pruning of the discriminator. Finally, they show state-of-the-art results on GAN compression through channel pruning.

**Strengths**
1. The paper is well-written and has cited relevant related work.

2. The work is well-motivated and novel. The authors answer some important questions about LTH based methods.

3. The experiments are extensive and help prove the authors' claims. I especially appreciate the attention to detail in the experiments, with various comparisions and ablation studies.

**Weaknesses and Clarifications**

From the given qualitative results, there does seem to be a loss in the finer features (edges and textures) upon sparsification. However, the FID scores do not seem to reflect this. Could the authors provide more visual results to analyse this?

---

> ### Author Response · Authors · 2020-11-21
> **Response to Reviewer #2 [Con 1]**
>
> Thank you for the detailed summary. We’re very glad you rate our work as well-motivated and novel.  Meanwhile, we found the set of perceptive questions you raised in your feedback very insightful, pushing us to further improve our paper.
>
> **[Con 1: Interpretation of Figures]**
>
> FID scores measure the similarity between two datasets, so FID scores might not well reflect the quality of individual images when some samples may lose finer features while some others gain better visual performance. We have provided more visualization results in the updated manuscript (Figure 11, Figure 12). Images generated by the best winning tickets seem to have more vivid colors and sharper edges, and images generated by tickets at the extreme sparsity have comparable qualitative results in spite of the checkerboard artifacts in some images. It is fair to note that the checkerboard artifacts broadly exist in generated from GAN models at all sparsity levels (even for unpruned GAN). Moreover, empirical evidence (Figure 3 in [1]) also reveals that checkerboard artifacts in some images do not necessarily result in higher FID scores on the whole datasets.
>
> [1] Fu, Yonggan, et al. “Autogan-distiller: Searching to compress generative adversarial networks.” ICML 2020.

---

### Official Review · AnonReviewer1 · 2020-10-29
**Empirical study of lottery ticket hypothesis on GANs is worth to share in this community**

**Rating:** 6
**Confidence:** 3

**Review:**

In this paper, the authors provide an empirical study on lottery ticket hypothesis on GANs. To do this, the authors use two GAN models and two datasets: SNGAN/CycleGAN and CIFAR-10/horse2zebra. Extensive experiments show that matching subnetworks can be found using unstructured magnitude pruning and channel pruning and they are transferrable to other tasks. The performance of subnetworks found is competitive and even surpasses state-of-the-art performance.

This paper is well-written and addresses an interesting problem in GANs compression. This paper first studies to verify if lottery ticket hypothesis works on GANs. Although the concept of LTH is not new, empirical verification on GANs is of value to this community. The experimental set-up including methods, datasets, and claims is solid and empirical results for the claims are convincing.

I am not familiar with the original work of lottery ticket hypothesis, but it is easy to understand the concept of it without reading the related works except for the following:
1. Is the iterative magnitude pruning (IMP) equivalent to the unstructured pruning described in p3?
2. What is the difference between matching subnetworks and winning tickets?
For self-contained paper and readers unfamiliar with LTH, it would be better if these terms are clarified in the revision.

The authors mainly use FID scores and additionally compare Inception Scores. It would be better to see if experiments on recent metrics (e.g. precision/recall) show the same result.

---

> ### Author Response · Authors · 2020-11-21
> **Response to Reviewer #1 [Cons 1-2]**
>
> Thank you for your appraisal of our manuscript. We truly value your positive comments on extensive and comprehensive empirical results and analyses. Below, we provide detailed responses to your concerns.
>
> **[Con 1: Clarification on Terms like IMP and Tickets]**
>
> We apologize that we did not make this clear and we have provided a clearer definition of these terms in the updated manuscript in Section 3. Here is the detailed clarification:
>
> - Unstructured pruning is a method that each of the weights can be individually set to zero [1], so the sparsity masks generated in this fashion can be in no structure. For comparison, channel pruning is a structured pruning method, because it will remove channels in kernels. The iterative magnitude pruning method is one of the unstructured pruning methods we choose to use, which iteratively prunes weights with the lowest magnitude [4] rather than pruning it only once to meet the sparsity requirement.
>
> - Matching subnetworks are subnetworks of full networks i) whose initialization is the weights of the full model at a certain training step and ii) that can reach a comparable result to the full networks when trained in isolation [2,3]. The winning tickets are matching subnetworks at initialization, i.e., training the winning ticket starting from its initialization (i.e., trained in isolation [2,3]), results in comparable performance to full unpruned models.
>
> [1] Cheng, Yu, et al. "A survey of model compression and acceleration for deep neural networks." ICLR 2019.
>
> [2] Chen, Tianlong, et al. "The lottery ticket hypothesis for pre-trained BERT networks." NeurIPS, 2020.
>
> [3] Frankle, Jonathan, et al. "Linear mode connectivity and the lottery ticket hypothesis.” ICML, 2020.
>
> [4] Renda, Alex, et al. "Comparing Rewinding and Fine-tuning in Neural Network Pruning." ICML, 2019.
>
> **[Con 2: New Evaluation Metrics]**
>
> Thank you for your precious suggestion. While FID and IS scores might be the several most widely used metrics to evaluate the performance of GANs, new metrics like precision and recall would provide a more comprehensive description. We will report these metrics (i.e., Precision, Recall,  F$_{1/8}$, and F$_8$ [5]) and graphs in the updated manuscript (Appendix B. 2, Figure 9 and Table 7).
>
> We also show the results here in Table S.1:
>
> Table S.1
>
> |          Models          | F$_8$ | F$_{1/8}$ | Sparsity |
> | :---------------------: | :---: | :-------: | :------: |
> |       Full Model        | 0.971 |   0.974   |  0.00%   |
> |  Best Winning Tickets   | 0.977 |   0.977   |  48.80%  |
> | Extreme Winning Tickets | 0.974 |   0.971   |  73.79%  |
>
>
> We observe from the above table that the best winning tickets perform superior to the full unpruned model regarding the two new evaluation metrics, ensuring the advantage of the best winning tickets found by the iterative pruning method. Meanwhile, the extreme winning tickets obtain comparable results to the full unpruned model. Overall, we find consistent observations across all metrics, including FID, IS scores, Precision, Recall, F$_{1/8}$, and F$_8$.
>
> [5] Sajjadi, Mehdi, et al. “Assessing generative models via precision and recall.” NeurIPS, 2018.

---

### Official Review · AnonReviewer4 · 2020-11-01

**Rating:** 6
**Confidence:** 4

**Review:**

In this paper, the ideas from lottery ticket hypothesis are evaluated in the case of SNGAN and CycleGAN.

The evaluations aim to validate the following empirical results with respect to the lottery ticket hypothesis i.e. the iterative magnitude pruning and channel pruning ideas work, discriminator can also be pruned in addition to generator, it outperforms other compression methods such as GAN slimming and that instead of re-initializing the weights of the masked network to the initialization, it could be reset to that from an earlier epoch.

While most of the ideas evaluated have already been proposed for general model compression, in this paper they are specifically evaluated for GANs.

The proposed work is useful in terms of the study and analysis of a particular approach from model compression to that of GANs, it would be useful to understand if the results are meaningful specially for the much larger GAN models such as Progressive GANs, BigGAN for high resolution images. This is because, compression is especially useful for the larger models and more so when one is generating high resolution images. Many of the early results on GANs trained on CIFAR 10/100 and SVHN do not hold for the larger GAN models.

To summarize the pros and cons of the paper are as follows:
Pros:
1) Evaluates various aspects of the Lottery ticket hypothesis for specific GAN models
2) Demonstrates results for both image to image and generation from latent cases and shows improvements over GAN slimming
3) Provides useful analysis for the lottery ticket hypothesis by analysing specific claims that are relevant

Cons:
1) Larger GAN models and larger datasets are not evaluated. The approach could be more meaningful for those models and datasets
2) The approach evaluates a specific lottery ticket hypothesis and does not provide new insights into the actual hypothesis itself. That is, by applying it to GANs we could only verify most of the already valid claims that were earlier proposed
3) It is not clear that the ideas proposed would generalize to other GAN models. There are a huge number of GAN models proposed in literature and validating the ideas on all would be infeasible. It would however be useful to provide some more results on a few more GAN models, particularly addressing the first point.

---
The response from the authors satisfactorily addresses several points that I raised. However, I am not fully convinced that the LTH on GANs has provided significant new insights. However, based on the response I am inclined to raise my score to above acceptance threshold and tend towards acceptance.

---

> ### Author Response · Authors · 2020-11-21
> **Response  to Reviewer #4 [Cons 1-3]**
>
> Thanks for your insightful comments. Below, we provide detailed responses to your concerns.
>
> **[Cons 1 & 3: More Experiments with Larger GAN Models and Datasets & More Experiments with Diverse GAN Structures.]**
>
> Thank you for your advice. We have conducted experiments of SAGAN [2] and DiffAugGAN (as large as BigGAN) [3] on CIFAR10 and, DCGAN [5], WGAN-GP [4] on Tiny ImageNet (i.e., the larger dataset with a higher image resolution).
>
> To verify the existence of lottery tickets in other diverse GANs models, we have conducted experiments on AutoGAN [9] (including three architecture variants), WGAN-GP [5], DCGAN [6], ACGAN [6], GGAN [7], ProjGAN [8], as well as the aforementioned SAGAN [2] and DiffAugGAN [3]. So far, we have updated several finished results (WGAN-GP [4], DCGAN [5], ACGAN [6], GGAN [7], ProjGAN [8], SAGAN [2], AutoGAN [9], DiffAugGAN [3]) to the manuscript. Detailed results are shown in Table S.2 and Table S.3.
>
> Table S.2: FID$_\mathrm{Full}$: The FID score of the baseline model. FID$_\mathrm{Best}$: The FID score of the best winning tickets. FID$_\mathrm{Extreme}$: The FID score of the extreme winning tickets. AutoGAN-A/B/C are three representative GAN architectures represented in the official repository (https://github.com/VITA-Group/AutoGAN).
>
> |   Model    | Benchmark | FID$_\mathrm{Full}$ (Sparsity) | FID$_\mathrm{Best}$ (Sparsity) | FID$_\mathrm{Extreme}$ (Sparsity) |
> | :--------: | :-------: | :----------------------------: | :----------------------------: | :-------------------------------: |
> |   DCGAN    | CIFAR-10  |           57.4 (0%)            |         49.31 (20.0%)          |           54.48 (67.2%)           |
> |  WGAN-GP   | CIFAR-10  |           19.2 (0%)            |         16.77 (36.0%)          |           17.28 (73.8%)           |
> |   ACGAN    | CIFAR-10  |           39.3 (0%)            |         31.45 (36.0%)          |           38.95 (79.0%)           |
> |    GGAN    | CIFAR-10  |           38.5 (0%)            |         33.42 (20.0%)          |           36.67 (48.8%)           |
> |  ProjGAN   | CIFAR-10  |           31.5 (0%)            |         28.19 (20.0%)          |           31.31 (67.2%)           |
> |   SAGAN    | CIFAR-10  |           14.7 (0%)            |         13.57  (20.0%)         |           14.68 (48.8%)           |
> | AutoGAN-A  | CIFAR-10  |           14.4 (0%)            |         14.04 (36.0%)          |           14.04 (36.0%)           |
> | AutoGAN-B  | CIFAR-10  |           14.6 (0%)            |         13.16 (20.0%)          |           14.20 (36.0%)           |
> | AutoGAN-C  | CIFAR-10  |           13.6 (0%)            |         13.41 (48.8%)          |           13.41 (48.8%)           |
> | DiffAugGAN | CIFAR-10  |           8.23 (0%)            |          8.05 (48.8%)          |           8.05 (48.8%)            |
>
> Table S.3: FID$_\mathrm{Full}$: The FID score of the baseline model. FID$_\mathrm{Best}$: The FID score of the best winning tickets. FID$_\mathrm{Extreme}$: The FID score of the extreme winning tickets.
>
> |   Model    |   Benchmark   | FID$_\mathrm{Full}$ | FID$_\mathrm{Best}$ (Sparsity) | FID$_\mathrm{Extreme}$ (Sparsity) |
> | :--------: | :-----------: | :-----------------: | :----------------------------: | :-------------------------------: |
> |   DCGAN    | Tiny ImageNet |       121.35        |         78.51 (36.0%)          |          114.00 (67.2%)           |
> |  WGAN-GP   | Tiny ImageNet |       211.77        |         194.72 (48.8%)         |          200.22 (67.2%)           |
>
> [2] Zhang, Han, et al. “Self-Attention Generative Adversarial Networks.” ICLR, 2019.
>
> [3] Zhao, Shengyu, et al. “Differentiable Augmentation for Data-Efficient GAN Training.” NeurIPS, 2020.
>
> [4] Gulrajani, Ishaan, et al. “Improved Training of Wasserstein GANs.” NeurIPS, 2017.
>
> [5] Radford, Alec, et al. “Unsupervised Representation Learning with Deep Convolutional Generative Adversarial Networks." ICLR, 2016.
>
> [6] Odena, Augustus, et al. “Conditional Image Synthesis with Auxiliary Classifier GANs.” ICML, 2017.
>
> [7] Lim, Jae Hyun, and Jong Chul Ye. “Geometric GAN.” arXiv, abs/1705.02894, 2017.
>
> [8] Miyato, Takeru, and Masanori Koyama. “cGANs with Projection Discriminator.” ICLR, 2018.
>
> [9] Gong, Xinyu, et al. “Autogan: Neural architecture search for generative adversarial networks.” ICCV, 2019.

---

> ### Author Response · Authors · 2020-11-21
> **(Continued) Response to Reviewer #4 [Con 2]**
>
> **[Con 2: More Insights]**
>
> We respectfully do not agree and point out insights in our work from the following three perspectives:
>
> - Existing works of the lottery ticket hypothesis (LTH) focus exclusively on discriminative networks. LTH on generative models has not been explored. Therefore, it remains a mystery whether there exist separate trainable winning tickets [1] in generative models since it has totally different properties compared with discriminative models. In this paper, we conducted extensive and systematic experiments on various models, which are shown in Figures 1, 4-8, and Table 1-5, and for the first time show the existence of lottery tickets in generative models. In addition, we provide comprehensive ablations and visualizations (Figure 2-3, 9-11) to reveal the properties of GAN tickets.
>
> - We also individually studied the effects of i) how to prune? (i.e., Only prune generators or discriminators? Prune both?) ii) how to initialize the subnetworks (i.e., Random initialization? Early rewind weights? Fully trained weights?), on the existence and quality of GAN tickets. As shown in Figure 4, we demonstrated that the different initialization of the discriminator will significantly influence the quality of sparse networks we found, while the pruning of the discriminator does not. These special scientific questions are exclusive and unique to GAN tickets, which are not covered by previous LTH literature.
>
> - Beyond this, we proposed another mechanism to utilize the trained weights of the dense network beyond weight rewinding, i.e. knowledge distillation. We used the knowledge distillation to assist in locating the matching subnetworks in GANs with superior performance. Combining knowledge distillation with LTH is also a novel contribution.
>
> [1] Frankle, Jonathan, and Michael Carbin. "The Lottery Ticket Hypothesis: Finding Sparse, Trainable Neural Networks." ICLR, 2018.

---

### Official Review · AnonReviewer5 · 2020-11-06
**Tend to Reject**

**Rating:** 6
**Confidence:** 4

**Review:**

GANs Can Play Lottery Tickets Too

Summary:

This paper investigates model pruning and the Lottery Ticket Hypothesis in the context of GAN training. The authors apply a set of pruning techniques to SN-GANs on CIFAR and CycleGANs and examine whether pruning masks can be applied to the models at their initialization state (or their state 20% into training), taking into consideration the GAN-specific choice of whether to independently prune G or to prune G and D jointly. The results suggest that the sparsity masks obtained from pruned networks can be applied to networks at initialization up to a given sparsity level while recovering similar or better performance on the target task.


My take:


This is an empirical paper which focuses its efforts on experimental design and elucidating some details of pruning in GAN training. While I think the experiments are reasonably well-designed for their intention, my primary concern is that I am not convinced that there’s any reason that lottery tickets should matter in this context at all, given that the process (a) requires one to train a model to completion in order to obtain the masks, and then (b) require access to the trained model for a distillation loss.

From my point of view, while it is interesting that one can use these masks on the networks in their init state, given that the actual goal of pruning is to obtain a slimmer model for inference (or to train models more quickly if pruning during training, which this technique does not permit), it is quite clear that standard pruning (pruning the pre-trained model rather than doing any sort of rewinding) is highly preferable in almost every way. That this result is downplayed in favor of highlighting that the authors have found lottery tickets is also concerning, albeit keeping with the general trend in the lottery ticket line of work which is more concerned with elucidating phenomenology than obtaining practical pruning or sparsity techniques. I feel strongly that the significance of this work will be diminished by this. On the other hand, I think that given that the intent of the paper is not as much to obtain the best possible sparsification technique as it is to investigate lottery tickets, this is perhaps okay, and I feel that the experimental design on this front is fairly strong and methodologically sound.

My other concern is that this paper’s clarity is poor. On the whole, the paper is not well presented, with poor or outright incorrect notation in many places. I struggled to read much of this paper due to this, and I feel that the paper’s impact would be strongly diminished due to this.

Overall, I would rate this paper about a 4.8. I will argue against acceptance, but I don’t think this is 100% a clear reject and depending on the opinions of the other reviewers I would not feel that accepting this paper was completely out of bounds.


Detailed and minor notes:



-Bibliography is inconsistently formatted: many of the references are missing information such as venue. If a paper is only on arXiv, the reference should indicate this. Note that one can use Scholar to automatically get the bibtex for a paper, and that one should do one’s best to get the most recent information on where a paper has landed. For example, the citation for the ResNet paper has no indication of the venue, when it was accepted by CVPR after being on arXiv first.


-” SNGAN with ResNet (He et al., 2015) is one of the most popular noise-to-image GAN network and has state-of-the-art performance on several datasets like CIFAR10.”

This is not true, SNGAN is far from SOTA on CIFAR. In general I don’t think we should care much about SOTA or “a few points below SOTA,” especially on datasets like CIFAR, but if the authors are specifically going to use the term “state-of-the-art” it is not acceptable to omit related work from the past three years. Please consider consulting paperswithcode for at least a fairly broad overview of recent literature that reports results on these datasets.


-“let g(x; θg) be the output of the generator network G “.

The authors re-use the notation “x” (typically used to refer to a sample from the dataset) to refer to the random noise latent input to G, instead of the standard practice “z.” This might be acceptable if the authors did not immediately then re-use “x” to refer to the input to the discriminator, and then again as the inputs to both G and D in the image-to-image task. Please correct this misuse of notation.

The authors incorrectly use the notation “. in R^d1” in multiple places to refer to multiple different things which certainly have different dimensionalities, and additionally mismatch the notation when referring to variables which *should* have the same dimensionality (such as referring to m in {0,1}^d, the mask for a parameter theta in R^d1. This is further exacerbated as the subscript “d” is used to refer to the discriminator parameters in e.g. theta_d. Please correct this misuse of notation.

-Figure 9: I can’t see any difference between the source image and the output of the full model, but I can see checkerboard artifacts in the highly sparse models. I found this figure confusing and out of place.

-In the background and related work, I feel that the authors omit (or do not make clear) the critical detail of the Lottery Ticket Hypothesis, which is that these subnetworks *exist at initialization* assuming one uses the same initialization weights (i.e. initializes uses the same random seed). It is also important to note that the Lottery Ticket Hypothesis in this form does not hold past the very small-scale tasks on which it was originally tested, which is what necessitates the “rewinding” technique (and, in this reviewer’s opinion, greatly weakens the hypothesis).

-“We use a benchmark pruning approach named Standard Pruning”

Is there a reference for standard pruning, or is this novel? If this is something the authors did not introduce, appropriate citations should be included.


-”In summary, initialization benefits finding winning tickets”
I was not able to understand this statement; please consider rewording it.

-It is not surprising that resetting the generator but not resetting the Discriminator fails to train (IMPFG in Figure 4), as Gs and Ds are basically always intrinsically paired in this manner.

=========================================================================================

Edit post Rebuttal:

Thank you to the authors for their in-depth response and the effort they put into responding to this review, and my apologies for not engaging during the discussion phase. I'll respond to several points with the goal of furthering the discussion to try and avoid unfairly "having the final say" when the authors cannot respond.

-"The biggest potential impact of our work is that it provides empirical evidence that lottery tickets exist in GANs"
This reviewer's opinion is that this is not at all surprising. While GANs do have unusual and unique training dynamics on a number of levels, and interact interestingly with many typical building blocks, many of the aspects of neural-network based GAN models (prunability, the relationship between signal propagation and trainability, model capacity, etc) are largely indistinguishable from those of more "typical" neural nets. Nonetheless I do agree that, if one holds the lottery ticket line of work to be important or relevant (which I personally do not, but I will withhold my bias and not "legislate from the bench" here), its extension into the realm of GANs does first require verification of its existence in this regime. I apologize to the authors if it seems like they're pushing against a brick wall here because this reviewer is not a disciple of the lottery ticket hypothesis--please note that to the best of my ability this review is meant to be calibrated around this bias and instead focus on the strength of the manuscript.

-Transfer between tasks: I appreciate the authors response on this point; transfer of masks between tasks (rather than pretrained weights) does indeed represent a different modality of transfer learning, and one which may be well-suited to GANs which are known to be difficult to re-train or fine-tune in many situations (although there is a growing body of work on this topic outside of the pruning/lottery ticket context).

-"Lottery tickets without early weight rewinding techniques can still hold to small-scale tasks. "
Yes, this is precisely the problem--lottery tickets without early weight rewinding *do not, in general, tend to hold on large scale tasks*. This again returns to my baseline issue (one of many) with the hypothesis in the first place, that it has to be modified to work on even models like VGG.

-on checkerboard artifacts: This reviewer is very well calibrated to viewing GAN samples, and holds that the checkerboard artifacts are vastly more visible in the sparsified models. I would encourage the authors to, in future work,  consider why this might be (what about sparse networks leads to increased checkerboard artifacts?) rather than seek to claim that the unpruned models are also checkerboard-y.

-Additional experiments: I commend the authors for repeating their experiments on a range of architectures, and agree that the improvements for the sparsified models are in support of their argumentation and conclusions.

-Thank you to the authors for updating their notation and bibliography.

On the whole, I still feel that this paper is borderline. While the author's responses are fairly strong in context, re-reading the manuscript, I still do not feel that the paper (which is most of what matters here) is especially strong. I am upgrading my score to around a 5.5 (which I will simply round up to a 6 on OpenReview). I think this paper is true borderline--I won't argue for its rejection, but I don't feel especially strongly in favor of it and cannot champion it.

---

> ### Author Response · Authors · 2020-11-21
> **Response to Reviewer #5 [Con 1]**
>
> We sincerely thank you for your precious suggestions and comments on our presentation and correctness. They greatly help us to improve our manuscript. We apologize for incorrect notations which have been updated in our new draft. Here are the detailed responses:
>
> **[Con 1: The Meaning of Exploring The Lottery Ticket Hypothesis (LTH) in GANs]**
>
> We believe that our research is scientific in nature, exploring and extending an empirical phenomenon in a new setting, i.e., generative adversarial networks. The biggest potential impact of our work is that it provides empirical evidence that lottery tickets exist in GANs, making it possible to reduce the cost of training or transferring large GAN models on different hardware platforms.
>
> **A. Efficiency Problems and Transfer Learning for GAN Tickets**
>
> - **It is worth mentioning that the cost of LTH can be amortized by transferring to other tasks, which has been studied both in LTH literature [9,12,15] and our paper’s Section 5.** Specifically, [9,12,15] provides an alternative way to transfer, i.e., the “transfer of initializations and masks” rather than the “transfer of learned representations” that previous works [13,14] focus on. In Section 5, we comprehensively study the transferability of GAN tickets and show that the extremely sparse GAN tickets we found on CIFAR-10 can transfer well to STL-10 without any pre-trained weights. It potentially provides empirical support for designing effective and original transfer learning mechanisms that is scarce [4,5] in GAN models.
> - Studying the winning tickets would help us to improve the theoretical understanding of neural networks, as suggested by [2]. For example, it was conjectured in [2] that the structure of winning tickets “encodes an inductive bias customized to the learning task”, which was discussed in [7].  The lottery ticket hypothesis also provided a complementary perspective on the relationship that “larger networks might explicitly contain simpler representations” [8]. Expanding the lottery ticket hypothesis to GANs can potentially provide theoretically sound insights into generative networks.
> - LTH needs “(a) requires one to train a model to completion in order to obtain the mask”, but it does not require “(b) require access to the trained model for a distillation loss.” (Refer to the answer bullet B blow). In this sense,  it is fair to note standard pruning and LTH have a similar training efficiency since standard pruning also needs pruning and then fine-tuning, and then repeat. The critical contribution and insight of LTH are that it implies a kind of subnetworks found in this way can train from the same initialization **in isolation** to **match or even surpass** the performance of full unpruned models. This finding tackled several previous obstacles like “training a pruned model from scratch performs worse than retraining a pruned model” [11]  and “it is better to retain the weights from the initial training phase for the connections that survived pruning than it is to re-initialize the pruned layers” [12]. Meanwhile, although it beyond our scope, there are several recent works [1] in LTH that can actually gain training efficiency by early stopping [1]. Specifically, Early Bird Tickets found in [1] can achieve 2.2 ~ 2.4x FLOPs reduction compared to the standard pruning process on discriminative models like ResNet and VGG. We will explore this kind of efficient winning tickets like [1] in GANs in the future, but its prerequisite - the existence of winning tickets - should be verified first, which is one of our contributions. It is also noteworthy that a main statement of the lottery ticket hypothesis is that the sparse network with proper initialization can reach comparable results to the original network when trained in isolation. If the hypothesis can be extended to GANs, then it implies that the inefficient process of training a GAN network is unnecessary “as one only needs to find a good small subnetwork and then train it separately” [6].
>
> **B. Distillation Loss is not Necessary and it is an Additional and Original Study for LTH**
>
> - For the knowledge distillation part, is an additional and original study. In other words, locating lottery tickets does not **require access to the trained model for a distillation loss**. We for the first time show that the knowledge distillation technique benefits to identify winning tickets with a larger sparsity (Figure 5). Even without using it, we can still find winning tickets of high quality in GAN models (Figure 4).
>
> **C. Sparification Techniques**
>
> - Thank you for valuing our experiment design as strong and methodologically sound. We respectfully point out that iterative magnitude pruning and channel pruning are two widely adopted and effective techniques to investigate the lottery ticket hypothesis. [1,2,3,6, 9,10,11,12]

---

> ### Author Response · Authors · 2020-11-21
> **(Continued) Response to Reviewer #5 [References for Con 1]**
>
> [1] You, Haoran, et al. "Drawing Early-Bird Tickets: Toward More Efficient Training of Deep Networks." ICLR, 2019.
>
> [2] Frankle, Jonathan, and Michael Carbin. "The Lottery Ticket Hypothesis: Finding Sparse, Trainable Neural Networks." ICLR, 2018.
>
> [3] Zhou, Hattie, et al. "Deconstructing lottery tickets: Zeros, signs, and the supermask." NeurIPS, 2019.
>
> [4] Frégier, Yaël, et al. “Mind2Mind : transfer learning for GANs. “ arXiv, abs/1906.11613, 2020
>
> [5] Wang, Yaxing et al. “Transferring GANs: generating images from limited data.” ECCV, 2018.
>
> [6] Malach, Eran, et al. "Proving the Lottery Ticket Hypothesis: Pruning is All You Need." ICML, 2020.
>
> [7] Cohen, Nadav, and Shashua, Amnon. “Inductive Bias of Deep Convolutional Networks through Pooling Geometry.” ICLR, 2017.
>
> [8] Zhou, Wenda, et al. “Compressibility and Generalization in Large-Scale Deep Learning.” arXiv, abs/1804.05862, 2018.
>
> [9] Chen, Tianlong, et al. "The lottery ticket hypothesis for pre-trained bert networks." NeurIPS, 2020.
>
> [10] Frankle, Jonathan, et al. "Linear mode connectivity and the lottery ticket hypothesis.” ICML, 2020.
>
> [11] Van Soelen, Ryan, and John W. Sheppard. "Using winning lottery tickets in transfer learning for convolutional neural networks." IJCNN, 2019.
>
> [12] Morcos, Ari, et al. "One ticket to win them all: generalizing lottery ticket initializations across datasets and optimizers." NeurIPS, 2019.
>
> [13] Molchanov, Pavlo, et al. "Pruning Convolutional Neural Networks for Resource Efficient Inference." ICLR, 2016.
>
> [14] Zhu, Michael H., and Suyog Gupta. "To Prune, or Not to Prune: Exploring the Efficacy of Pruning for Model Compression." ICLR Workshop, 2018.
>
> [15] Mehta, Rahul. "Sparse transfer learning via winning lottery tickets." arXiv,  abs/1905.07785, 2019.

---

> ### Author Response · Authors · 2020-11-21
> **(Continued) Response to Reviewer #5 [Cons 2&3]**
>
> **[Con 2: Experiments on More Recent GANs]**
>
> We apologize for our inconsideration. We fixed this statement in our updated manuscript. We also have conducted new experiments on multiple GAN models, including GGAN [22], DCGAN [16], WGAN [17], ACGAN [18], ProjGAN [23], SAGAN [19], DiffAugGAN [20], AutoGAN [21], among which the DiffAugGAN [20] is a near-SOTA method on CIFAR-10 (https://paperswithcode.com/sota/image-generation-on-cifar-10), to show the validity of our methods. The results are shown in Section 6, Table 5, and Table 6. We are able to find winning tickets across all GAN networks. Detailed results are shown in Table S.4 and Table S.5.
>
>
> Table S.4: FID$_\mathrm{Full}$: The FID score of the baseline model. FID$_\mathrm{Best}$: The FID score of the best winning tickets. FID$_\mathrm{Extreme}$: The FID score of the extreme winning tickets. AutoGAN-A/B/C are three representative GAN architectures represented in the official repository (https://github.com/VITA-Group/AutoGAN).
>
> |Model|Benchmark| FID$_\mathrm{Full}$ (Sparsity)| FID$_\mathrm{Best}$ (Sparsity)| FID$_\mathrm{Extreme}$ (Sparsity)|
> | :-: | :-: | :-: | :-: | :-: |
> |   DCGAN    | CIFAR-10  |57.4 (0%) |         49.31 (20.0%)          |           54.48 (67.2%)           |
> |  WGAN-GP   | CIFAR-10  | 19.2 (0%) |         16.77 (36.0%)          |           17.28 (73.8%)           |
> |   ACGAN    | CIFAR-10  |39.3 (0%) |         31.45 (36.0%)          |           38.95 (79.0%)           |
> |    GGAN    | CIFAR-10  | 38.5 (0%) |         33.42 (20.0%)          |           36.67 (48.8%)           |
> |  ProjGAN   | CIFAR-10  | 31.5 (0%) |         28.19 (20.0%)          |           31.31 (67.2%)           |
> |   SAGAN    | CIFAR-10  |14.7 (0%) |         13.57  (20.0%)         |           14.68 (48.8%)           |
> | AutoGAN-A  | CIFAR-10  | 14.4 (0%) |         14.04 (36.0%)          |           14.04 (36.0%)           |
> | AutoGAN-B  | CIFAR-10  | 14.6 (0%) |         13.16 (20.0%)          |           14.20 (36.0%)           |
> | AutoGAN-C  | CIFAR-10  | 13.6 (0%) |         13.41 (48.8%)          |           13.41 (48.8%)           |
> | DiffAugGAN | CIFAR-10  | 8.23 (0%) |          8.05 (48.8%)          |           8.05 (48.8%)            |
>
> Table S.5: FID$_\mathrm{Full}$: The FID score of the baseline model. FID$_\mathrm{Best}$: The FID score of the best winning tickets. FID$_\mathrm{Extreme}$: The FID score of the extreme winning tickets.
>
> |   Model    |   Benchmark   | FID$_\mathrm{Full}$ | FID$_\mathrm{Best}$ (Sparsity) | FID$_\mathrm{Extreme}$ (Sparsity) |
> | :--------: | :-----------: | :-----------------: | :----------------------------: | :-------------------------------: |
> |   DCGAN    | Tiny ImageNet |       121.35        |         78.51 (36.0%)          |          114.00 (67.2%)           |
> |  WGAN-GP   | Tiny ImageNet |       211.77        |         194.72 (48.8%)         |          200.22 (67.2%)           |
>
> [16] Radford, Alec, et al. “Unsupervised Representation Learning with Deep Convolutional Generative Adversarial Networks.” ICLR, 2016.
>
> [17] Gulrajani, Ishaan, et al. “Improved Training of Wasserstein GANs.” NeurIPS, 2017.
>
> [18] Odena, Augustus, et al. “Conditional Image Synthesis with Auxiliary Classifier GANs.” ICML, 2017.
>
> [19] Zhang, Han, et al. “Self-Attention Generative Adversarial Networks.” ICLR, 2019.
>
> [20] Zhao, Shengyu, et al. “Differentiable Augmentation for Data-Efficient GAN Training.” NeurIPS, 2020.
>
> [21] Gong, Xinyu, et al. “Autogan: Neural architecture search for generative adversarial networks.” ICCV, 2019.
>
> [22]  Lim, Jae Hyun and Ye, Jong Chul. “Geometric GAN.” arXiv, abs/1705.02894, 2017.
>
> [23] Miyato, Takeru, and Masanori Koyama. “cGANs with Projection Discriminator.” ICLR, 2018.
>
> **[Con 3: Checkerboard Artifacts in Figures]**
>
> We admit that checkerboard artifacts exist in images generated by highly sparse models, but it is fair to note that these patterns can also be observed in generated images from full unpruned models (2nd row, 2nd column, Figure 2). The same observation exists in Figure 4 of [24] too. We have also included more figures (Figure 11, Figure 12) to show the qualitative results in the updated manuscript. The images we selected following the setting of Figure 3 in [25]. For the summer2winter dataset, the checkerboard artifacts can be observed from the two figures of the full models in the first row of Figure 12. For more comparison, we provide two extra generation results in the second row of Figure 11 and Figure 12. The checkerboard artifacts exist in GAN models across all sparsity levels, and the generated images appear to have a similar visual quality. The images generated by GAN winning tickets seem to have more vivid colors and more details.
>
> [24] Shen, Zengming, et al. "One-to-one Mapping for Unpaired Image-to-image Translation." WACV, 2020
>
> [25] Shu, Han, et al. "Co-Evolutionary Compression for Unpaired Image Translation." ICCV, 2019.

---

> ### Author Response · Authors · 2020-11-21
> **(Continued) Response to Reviewer #5 [Cons 4-7]**
>
> **[Con 4: The Same Random Initialization on LTH]**
>
> We will enrich our description of the lottery ticket hypothesis in related works, change the term “at the initialization” to “at the same random initialization”.
>
> We respectfully disagree with your statement on the ‘rewinding weakens the lottery ticket hypothesis’ in the following two cases:
>
> - *If the “rewinding” refers to the early weight rewinding techniques:* Lottery tickets without early weight rewinding techniques can still hold to small-scale tasks. It is not only verified by us in Figure 1 and Figure 8 but also covered in the previous literature [1,9]. Early weight rewinding is an effective and complementary technique to scale up the lottery ticket hypothesis and the matching subnetworks, as stated by [10].
>
> - *If the “rewinding” refers to resetting weight to the same initialization:* Rewinding to the same initialization is one of the critical properties of the lottery ticket hypothesis. It implies that the subnetwork found in this way can train from the same initialization in isolation to match or even surpass the performance of full unpruned models. This is one of the most impressive and major contributions of the lottery ticket hypothesis since it tackled some previous obstacles like “training a pruned model from scratch performs worse than retraining a pruned model” [26]  and “it is better to retain the weights from the initial training phase for the connections that survived pruning than it is to re-initialize the pruned layers” [27].
>
> [26] Li, Hao, et al. “Pruning Filters for Efficient ConvNet.”, ICLR, 2017.
>
> [27] Han, Song, et al. "Learning both weights and connections for efficient neural network." NeurIPS, 2015.
>
> **[Con 5: Intrinsically Paired Relationships in GAN Models]**
>
> In the previous handful of GAN compression studies [24,28,29], people usually only prune the generative model since they focus on the inference stage where it does not need the discriminators. Surprisingly, although the paired architecture relationship between the discriminator and the generator seems to have been destroyed under this context, only pruning the generator still works well [24,28,29]. In our case, the situation is more complicated since the discriminators need to be involved in the process of locating GAN tickets.
>
> Particularly, we investigated two different kinds of imparity, i.e., imparity in structure (pruned generators with un-pruned discriminators) and imparity in weights (resetted generator weights and un-resetted discriminator weights). It remains unknown whether the broken paired relationship is beneficial or harmful for the existence and quality of identified GAN tickets. The comprehensive studies are presented in Figure 4. We think these investigations and analyses are crucial and novel, especially for the LTH community.
>
> [28] Wang, Haotao, et al. “GAN Slimming: All-in-One GAN Compression by A Unified Optimization Framework.” ECCV, 2020.
>
> [29] Li, Muyang, et al. "Gan compression: Efficient architectures for interactive conditional gans." CVPR, 2020.
>
> **[Con 6: Standard Pruning, and Rewording Descriptions]**
>
> We have also added the citation for the standard pruning method ([1]) and reworded the sentence for better understanding to “In summary, different initialization of the discriminator will significantly influence the existence and quality of winning tickets in GAN models. ”
>
> **[Con 7: Wrong Notions and Bibliography]**
>
> Thank you for pointing out incorrect notions and incomplete bibliography. We made corrections and marked them in the updated version of the manuscript.
>
> We followed your suggestion to change the input of generators from $x$ to $z$, and change the notions of dimension to $d_z$ and $d_\theta$ to avoid the abuse of $d$. Also, we have updated all the outdated and wrong entries in the manuscript.

---

### Author Response · Authors · 2020-11-21
**General Responses**

We would like to thank all reviewers for providing many useful feedbacks. We sincerely apologize for this delayed response, since the author team had been extremely occupied by another deadline just passed.

Here is a summary of our updates:

- Below we address all questions raised and provide point-to-point responses.

- As mentioned by reviewer #5, #4, and #1, new experiment results with extra evaluation metrics, larger datasets, and larger and diverse GAN models, are provided, including WGAN-GP, DCGAN, ACGAN, GGAN, ProjGAN, SAGAN, AutoGAN (three variants), DiffAugGAN on CIFAR-10, and DCGAN, WGAN-GP on Tiny ImageNet.  We will keep updating more before the end of the rebuttal period.

- The modified nine pages draft is updated. We will keep updating for better and clear readability (we thanks again for R5’s precious comments on our presentation and notation correctness).

- For reproducibility, both training and evaluation codes for GAN tickets are provided as additional supplementary material. And identified winning tickets in SNGAN can be found at https://www.dropbox.com/s/xqvmxt7fl0mxr7u/best_winning_tickets.pth?dl=0 (the best winning tickets) and https://www.dropbox.com/s/l27iaolkwhk4xrp/extreme_winning_tickets.pth?dl=0 (the extreme winning tickets).

We hope our responses, although coming a bit late in this time window (we apologize again), have clarified all confusion and could help reviewers more fairly and positively assess our work. We thank all reviewers’ time again.

---

### Decision · Program_Chairs · 2021-01-07
**Final Decision**

**Decision:**

Accept (Poster)

**Comment:**

Three reviewers recommended accept or weak accept. There are some concerns on the novelty of this approach since this work mainly validates that lottery tickets can be found on GANs, which seems like applying an existing idea to a new problem. Nevertheless, there are two reasons that such an effort is interesting: first, the implementation of this idea may be harder than it seems; second, it was not known a priori whether lottery tickets would exist for GANs due to the significantly different optimization problem (game instead of minimization). I think this work will be of interest to the community, and recommend acceptance.